# A2R: An Asymmetric Two-Stage Reasoning Framework for Parallel Reasoning

## Abstract

Recent Large Reasoning Models have achieved significant improvements in complex task-solving capabilities by allocating more computation at the inference stage with a "thinking longer" paradigm. Even as the foundational reasoning capabilities of models advance rapidly, the persistent gap between a model's performance in a single attempt and its latent potential, often revealed only across multiple solution paths, starkly highlights the disparity between its realized and inherent capabilities. To address this, we present A2R, an Asymmetric Two-Stage Reasoning framework designed to explicitly bridge the gap between a model's potential and its actual performance. In this framework, an "explorer" model first generates potential solutions in parallel through repeated sampling. Subsequently, a "synthesizer" model integrates these references for a more refined, second stage of reasoning. This two-stage process allows computation to be scaled orthogonally to existing sequential methods. Our work makes two key innovations: **First**, we present **A2R** as a plug-and-play parallel reasoning framework that explicitly enhances a model's capabilities on complex questions. For example, using our framework, the Qwen3-8B-distill model achieves a 75% performance improvement compared to its self-consistency baseline. **Second**, through a systematic analysis of the explorer and synthesizer roles, we identify an effective asymmetric scaling paradigm. This insight leads to **A2R-Efficient**, a "small-to-big" variant that combines a Qwen3-4B explorer with a Qwen3-8B synthesizer. This configuration surpasses the average performance of a monolithic Qwen3-32B model at a nearly 30% lower cost. Collectively, these results show that A2R is not only a performance-boosting framework but also an efficient and practical solution for real-world applications.

## 1 Introduction

Large Language Models (LLMs) have achieved remarkable progress in solving complex reasoning tasks, driven by advances in model scaling, data quality, and both training and inference-time techniques (Brown et al., 2020; Chowdhery et al., 2023; OpenAI, 2023; Touvron et al., 2023; DeepSeek-AI et al., 2025; Team et al., 2025). Among these, inference-time methods such as Chain-of-Thought prompting (Wei et al., 2022) greatly enhance reasoning by encouraging models to generate explicit intermediate steps, thereby improving performance on complex tasks. Building on this, multi-path approaches such as self-consistency (Wang et al., 2023) and best-of-N (Brown et al., 2024a) decoding further boost accuracy by sampling diverse reasoning trajectories. Early studies introduce self-reflection and self-correction mechanisms (Madaan et al., 2023; Shinn et al., 2023; Kumar et al., 2025), which improve robustness by allowing LLMs to critique and revise their own reasoning steps.

While these advances have elevated model performance, a substantial gap remains between single-pass and multi-pass reasoning. Across diverse benchmarks, models consistently achieve much higher pass@K scores when multiple reasoning paths are sampled, underscoring the limitations of single-pass inference. For example, math reasoning tasks often show gains of 15–20 points when scaling from pass@1 to pass@8. This gap reflects a core limitation of the prevailing single-path decoding paradigm: once a suboptimal step is taken early in the reasoning process, the entire trajectory can be irreversibly diverted—a phenomenon termed the 'prefix trap' (Luo et al., 2025).

In this work, we explore the parallel allocation of inference computation—a dimension of improvement that is orthogonal to both parameter scaling and sequential inference-time scaling, making

it complementary to existing approaches. While previous inference-time scaling techniques, such as self-consistency (Wang et al., 2023), can improve model performance by aggregating outcomes from multiple independently generated reasoning paths, they suffer from two fundamental limitations. First, the independent generation of candidates leads to redundant computation and prevents information sharing across paths. Second, the aggregation stage is purely selective—typically relying on simple voting or ranking Cobbe et al. (2021); Uesato et al. (2022b); Aggarwal et al. (2023); Wang et al. (2024), without performing any additional reasoning or integrating insights from the complete set of reasoning chains. Consequently, it remains an open question what characteristics would define a more effective model for this aggregation and synthesis role.

We propose **Asymmetric Two-Stage Reasoning (A2R)**, a novel framework that enhances language model reasoning by decoupling inference into two complementary phases: a divergent exploration stage that produces diverse reasoning paths, and a convergent synthesis stage that integrates them. Unlike prior parallel inference approaches that aggregate answers through simple voting or selection, **A2R** introduces a Synthesizer model that performs generative re-reasoning over the full set of reasoning chains. This process enables the model to form a holistic view of the candidate solutions, identify consistent evidence, and synthesize a more accurate and robust final answer.

To assess the effectiveness of **A2R**, we first apply it to a setting where the same model is used in both stages. On complex reasoning benchmarks like AIME 2024, AIME (AIME2024, 2024; AIME2025, 2025), and BeyondAIME (BeyondAIME, 2025), this two-stage process delivers substantial gains; for example, employing Qwen3-8B-distill as both explorer and synthesizer with four reasoning paths achieves a 75% relative performance increase over the majority voting baseline, demonstrating the benefit of structured re-reasoning without any change in model size or training.

The significant improvements in the symmetric setup motivated us to dissect the framework and uncover the key drivers of performance. We conduct a systematic analysis of model capabilities within their respective A2R roles. Our results reveal a strong positive correlation between synthesizer capacity and performance gains: stronger synthesizers consistently deliver better outcomes. This shows that the synthesizer must exceed the explorer in reasoning ability, and that treating it as a mere router is both insufficient and suboptimal.

Motivated by these findings, we introduce **A2R-Efficient**, an asymmetric architecture for parallel reasoning. In this design, a smaller Explorer generates diverse reasoning paths, while a larger Synthesizer performs a final re-reasoning step to produce a consolidated answer. The Synthesizer is further fine-tuned with reinforcement learning, enabling it to critically evaluate candidate paths and generate more reliable outputs. This asymmetric configuration achieves accuracy comparable to a much larger single model while reducing computational cost by about 30%. Overall, A2R demonstrates an efficient strategy for parallel inference: minimize the expense of exploration and allocate greater capacity to synthesis to maximize performance.

**Our main contributions** are as follows. **First**, we propose **A2R**, a plug-and-play framework that decouples reasoning into a parallel exploration phase and a synthesis phase. Unlike prior methods that rely on passive selection, A2R introduces explicit generative re-reasoning over complete reasoning chains, substantially narrowing the gap between single-pass performance and a model's latent reasoning potential. **Second**, through a systematic analysis of the framework's internal roles, we identify an effective asymmetric scaling paradigm: the synthesizer's capacity is the critical bottleneck and the primary driver of performance. Building on this insight, we introduce **A2R-Efficient**, which uses a lightweight Explorer with a stronger Synthesizer. This configuration matches the performance of a much larger monolithic model while reducing computational cost by about 30%. Together, these contributions establish A2R as a principled and efficient strategy for unlocking the full reasoning potential of LLMs through coordinated parallel inference.

## 2 RELATED WORK

**Test-time scaling** Increasing computational overhead at test-time to boost the performance of LLMs on complex tasks has become a widespread and effective research paradigm. Chain-of-Thought (Wei et al., 2022) prompting pioneered the use of step-by-step reasoning, representing a pivotal shift away from intuitive System1(Li et al., 2025) processes. Building on this, tree- and graph-based (Yao et al., 2023; Besta et al., 2024) methods use explicit backtracking and branching to navigate a landscape of potential solutions, moving beyond a single path to further expand the

computational budget. A recent breakthrough in tackling long-horizon reasoning involves Reinforcement Learning with Variable Reward (RLVR), an approach pioneered by models like OpenAI-O1, DeepSeek-R1 (OpenAI, 2024; DeepSeek-AI et al., 2025) that signals the arrival of System 2 capabilities. However, a prefix trap (Luo et al., 2025) will still constrain the model's performance when it begins with a poor start, though it has the capability to self-correct, which seems a natural trouble in causal language models. Furthermore, although models enhanced by advanced RLVR algorithms achieve significant performance, a persistent gap between their pass@1 and pass@k scores remains, irrespective of their overall strength. Therefore, we aims to establish a novel paradigm for test-time computation to reconcile a model's potential capabilities with its observable performance.

**Parallel reasoning** As a foundational parallel reasoning paradigm, self-consistency (Wang et al., 2023) methods concurrently generate multiple responses and select the final output through voting. Another line of research involves methods like Process Reward Models and Outcome Reward Models (Uesato et al., 2022a), which utilize an external judge to identify the best answer through a process known as Best-of-N (Brown et al., 2024b) sampling. More recently, the research community has seen the emergence of sophisticated parallel reasoning frameworks like Adaptive Parallel Reasoning (Pan et al., 2025) and Multiverse (Yang et al., 2025), which explore more efficient computation structures and incorporate advanced engineering optimizations. Closely related to our work, the Sample Set Aggregator (SSA) (Qi et al., 2025) also utilizes a separate model to aggregate multiple sampled outputs rather than relying on simple voting. However, our work differs from SSA in two critical aspects. First, our A2R framework analyzes resource allocation between the exploration and synthesis stages from a computational cost perspective, leading to our proposed efficient, asymmetric "small explorer, large synthesizer" architecture. Second, we conduct a systematic analysis of the second-stage Synthesizer model, revealing that its intrinsic reasoning capacity is the key determinant of the final performance upper bound—a factor not fully explored in prior work.

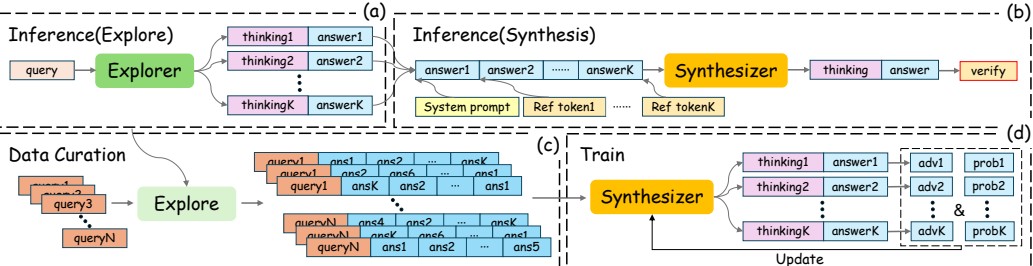

Figure 1: Figure 1: Overview of the A2R framework.(a) Generating multiple reasoning traces and candidate answers in parallel.(b) Integrating traces and performing a re-reasoning step to produce a solution.(c) Collecting reasoning paths from the Explorer for training data curation.(d) Fine-tuning the Synthesizer with reinforcement learning on the collected paths.

## 3 METHOD

In this section, we introduce in detailed of **Asymmetric Two-Stage Reasoning (A2R)** framework. A2R consists of two complementary stages: an exploration stage, where the Explorer generates multiple independent reasoning paths (Section 3.1), and a synthesis stage, where the Synthesizer integrates and re-reasons over these paths to produce a consolidated solution (Section 3.2). To further improve synthesis, we fine-tune the Synthesizer with reinforcement learning (Section 3.3). An overview of the A2R architecture is provided in Figure 1.

### 3.1 EXPLORER

The first stage of our framework is designed for exploration. Given a query $Q$, the Explorer model $M_E$ generates $N$ diverse reasoning paths in parallel. Each path $P_i$ consists of a detailed reasoning trace $T_i$ and a concise answer component $A_i$:

$$P_i = (T_i, A_i) \quad \text{for } i = 1, \dots, N$$

To manage the context length limitations of the subsequent stage, we construct a reference context $R_{\text{ref}}$, by concatenating only the answer components $A_i$, from each of the $N$ paths. It's important to note that each $A_i$ is not just a final numerical or single-word answer but is itself a concise chain of thought that shows how the answer was derived.

$$R_{\text{ref}} = \text{concat}(A_1, A_2, \ldots, A_N)$$

The purpose of this stage is to cast a wide net, capturing multiple potential lines of reasoning. This rich, multi-faceted context serves as the foundation for the next stage.

## 3.2 SYNTHESIZER

The second stage performs synthesis. Here, the Synthesizer Model $M_S$ is prompted with a composite query $Q'$—formed by combining the original query $Q$ with the reference context $R_{\text{ref}}$ from the Explorer—to carry out a consolidated re-reasoning step.

---

**SYNTHESIZER PROMPT TEMPLATE**

**Instruction**: You can solve the problem using the provided references, or you can choose to find a new solution. The final answer should be placed in boxed{}.
**Query**: Original Query $Q$
**Reference**: $< reference1 > A_1 < /reference1 >< reference2 > A_2 < /reference2 >$ $\cdots < referenceN > A_N < /referenceN >$

---

Unlike simple aggregation methods like majority voting that only consider the final outcomes, the Synthesizer leverages the full reasoning context of the candidate answers. It performs a generative synthesis, allowing it to identify correct steps from flawed paths, correct errors, and produce a more accurate reasoning chain and final answer $A_{final}$.

## 3.3 OPTIMIZING SYNTHESIS WITH REINFORCEMENT LEARNING

To enhance the Synthesizer's reasoning ability and its capacity to better critique reference prompt, we employ reinforcement learning (RL) and formulate the task as a policy optimization problem. Our approach is based on the GRPO (Shao et al., 2024) algorithm, which we adapt in several key aspects. Following the DAPO (Yu et al., 2025) framework, we apply a token-level policy gradient loss, a length-aware penalty, and dynamic sampling. The latter technique is particularly critical for addressing the frequent "zero advantage" scenarios that arise from our paradigm's high reward-acceptance rate. Notably, we replace the critic's "Clip-Higher" mechanism from DAPO with a symmetric, fixed threshold for both low and high clipping bounds. As we found that for smaller models, such as Qwen-4B and Qwen-8B, a lower high-clipping value significantly improves training stability. Furthermore, we have introduced several key modifications to ensure our training process remains stable:

- **On-Policy Updates**: To stabilize the training process, we set the $train\_batch\_size$ and $train\_mini\_batch\_size$ to be identical. This adjustment ensures that our updates are performed in a fully on-policy manner.

- **Controlled Temperature for Entropy Control**: We observed that a relatively controlled temperature of 0.7 helps maintain the entropy of the policy within a stable region. Our experiments revealed that without this constraint, the entropy can increase exponentially once it surpasses a certain threshold, leading to training instability.

The learning process is guided by a simple binary reward based on the final answer's correctness:

$$R(\hat{y}, y) = \begin{cases} 1, & \text{if is\_equivalent}(\hat{y}, y) \\ 0, & \text{otherwise} \end{cases}$$

The final policy is optimized using the following objective function:

$$J(\theta) = \mathbb{E}_{(q, r_{\text{ref}}, a) \sim D, R_{ref}, \{o_i\}_{i=1}^G \sim \pi_{\theta_{\text{old}}}(\cdot|q, r_{\text{ref}})}$$

$$\left[ \frac{1}{\sum_{i=1}^G |o_i|} \sum_{i=1}^G \sum_{t=1}^{|o_i|} \min \left( r_{i,t}(\theta) \hat{A}_{i,t}, \text{clip}\left(r_{i,t}(\theta), 1 - \epsilon, 1 + \epsilon\right) \hat{A}_{i,t} \right) \right] \quad (1)$$

$$\text{s.t.} \quad 0 < |\{o_i \mid \text{is\_equivalent}(a, o_i)\}| < G,$$

where $r_{i,t}(\theta)$ is the policy probability ratio and $\hat{A}_{i,t}$ is the standardized advantage, calculated as:

$$r_{i,t}(\theta) = \frac{\pi_\theta(o_{i,t}|q, r_{\text{ref}}, o_{i,<t})}{\pi_{\theta_{\text{old}}}(o_{i,t}|q, r_{\text{ref}}, o_{i,<t})}, \quad \hat{A}_{i,t} = \frac{R_i - \text{mean}(\{R_i\}_{i=1}^G)}{\text{std}(\{R_i\}_{i=1}^G)}$$

## 4 EXPERIMENT

We evaluate the proposed **A2R** framework through a series of experiments. Our study begins with a broad evaluation across models of varying sizes and origins, establishing the generality of **A2R**. We then perform a systematic analysis to isolate the role of the Synthesizer's reasoning capacity, identifying it as the critical driver of performance. Finally, building on this insight, we investigate an asymmetric configuration that pairs a smaller Explorer with a larger Synthesizer, showing that it can rival the performance of a much larger monolithic model with significantly improved efficiency.

### 4.1 EXPERIMENTAL SETTINGS

**Data.** Our reinforcement learning adopts an off-policy strategy for efficiency. Since the Explorer model is frozen during training, we decouple it from the RL loop and perform its inference externally to construct the training dataset. Concretely, we curated a set of approximately 10k challenging queries from the Skywork-OR1 dataset (He et al., 2025), focusing on complex problems where the **A2R** framework shows the most room for improvement. For each query, the Explorer generated up to 16 diverse and valid candidate solutions via repeated sampling. To further enrich the data, we introduced diversity by shuffling reasoning paths and by dynamically sampling different target answers for the same query during training. This strategy yields a large and diverse dataset cost-effectively, while avoiding unnecessary recomputation of Explorer outputs during training.

**Training.** To ensure the generality of our findings, our experiments were conducted on various models from the Qwen series and a Deepseek-distilled model and all the experiments were conducted on verl (Sheng et al., 2024) with a maximum sequence length of 8,192 tokens and a training batch size of 32. All models were trained on the curated dataset until convergence. To assess the performance of our framework, we evaluate it on a suite of highly challenging mathematical reasoning benchmarks derived from real-world competitions: AIME 2024, AIME 2025, and Beyond AIME. To ensure stable and reproducible results, we report the standard pass@1 accuracy for all benchmarks, averaged over 16 independent evaluation runs.

### 4.2 MAIN RESULTS

As presented in Table 1, our **A2R** framework delivers a significant performance uplift across three challenging benchmarks, consistently outperforming both standard single-pass inference (Pass@1) and the strong self-consistency (Cons@N) baseline across all tested models and path counts (N). For the state-of-the-art Qwen3-32B model at N=4, A2R achieves an average score of 74.62, surpassing Cons@N by over a full percentage point and the Pass@1 baseline by nearly 7 points. This robust outperformance validates our core hypothesis: A2R's generative synthesis, which actively re-reasons over diverse evidence, is a fundamentally more powerful approach than the passive, selective aggregation of majority voting.

| Benchmarks | Models | Pass@1 | N=4 | | | N=8 | | | N=12 | | |
|---|---|---|---|---|---|---|---|---|---|---|---|
| | | | Cons@N | Pass@N | A2R | Cons@N | Pass@N | A2R | Cons@N | Pass@N | A2R |
| AIME 2024 | Qwen3-4B | 74.63 | 79.38 | 81.44 | 80.03 | 80.05 | 82.68 | 80.50 | 80.05 | 83.61 | 80.92 |
| | Qwen3-8B | 75.30 | 79.34 | 82.76 | 79.88 | 80.47 | 84.58 | 80.55 | 80.45 | 85.74 | 80.52 |
| | Qwen3-32B | 81.15 | 84.74 | 88.75 | 86.89 | 85.37 | 91.21 | 86.76 | 85.66 | 92.29 | 86.85 |
| | Qwen3-8B-Distill | 82.30 | 84.21 | 87.84 | 86.78 | 84.38 | 90.18 | 87.17 | 84.38 | 91.53 | 87.42 |
| AIME 2025 | Qwen3-4B | 65.83 | 73.70 | 80.47 | 74.44 | 76.48 | 83.90 | 76.15 | 77.50 | 85.47 | 75.44 |
| | Qwen3-8B | 68.67 | 74.84 | 80.48 | 77.61 | 77.38 | 83.18 | 77.40 | 78.16 | 84.56 | 77.63 |
| | Qwen3-32B | 73.32 | 78.45 | 84.14 | 81.45 | 80.33 | 86.07 | 81.48 | 80.58 | 87.22 | 80.95 |
| | Qwen3-8B-Distill | 74.93 | 79.12 | 84.62 | 82.05 | 81.57 | 86.69 | 83.13 | 82.38 | 87.94 | 83.85 |
| BeyondAIME | Qwen3-4B | 42.10 | 45.75 | 55.39 | 48.94 | 47.96 | 59.52 | 49.95 | 48.78 | 61.47 | 50.13 |
| | Qwen3-8B | 43.94 | 47.24 | 57.13 | 50.07 | 48.94 | 62.07 | 51.31 | 49.41 | 64.51 | 52.15 |
| | Qwen3-32B | 48.80 | 52.39 | 61.66 | 55.52 | 54.51 | 66.31 | 55.47 | 55.09 | 68.64 | 55.23 |
| | Qwen3-8B-Distill | 52.93 | 57.47 | 65.08 | 60.04 | 59.89 | 68.85 | 61.28 | 60.78 | 70.58 | 62.25 |
| Average | Qwen3-4B | 60.85 | 66.28 | 72.43 | 67.80 | 68.16 | 75.37 | 68.87 | 68.78 | 76.85 | 68.83 |
| | Qwen3-8B | 62.64 | 67.14 | 73.46 | 69.19 | 68.93 | 76.61 | 69.75 | 69.34 | 78.27 | 70.10 |
| | Qwen3-32B | 67.76 | 71.86 | 78.18 | 74.62 | 73.40 | 81.20 | 74.57 | 73.78 | 82.72 | 74.34 |
| | Qwen3-8B-Distill | 70.05 | 73.60 | 79.18 | 76.29 | 75.28 | 81.91 | 77.19 | 75.85 | 83.35 | 77.84 |

Table 1: Detailed performance metrics of the A2R framework, including Pass@1, Cons@K, Pass@K, and the final A2R score across different numbers of exploration paths.

Furthermore, the results unveil a compelling scaling trend: the performance advantage of A2R over self-consistency becomes more pronounced as the base model's capability increases. At N=4, for example, A2R's average improvement over Cons@N is +1.52 points for Qwen3-4B, but this advantage expands to +2.76 points for the much stronger Qwen3-32B model. This indicates that A2R is particularly effective at unlocking the enhanced reasoning abilities of more powerful models. We also observe that A2R's absolute performance scales with N, though its relative gain over Cons@N is most significant at practical, lower values of N. A systematic analysis of these scaling dynamics is provided in the following section.

### 4.3 DEEP ANALYSIS: THE ROLE OF THE SYNTHESIZER

In this section, we present a deep analysis to elucidate the ideal characteristics of the Synthesizer model. This analysis, with results summarized in Table 2, comprises three key experiments designed to probe the relationship between model capability and the A2R framework's performance.

**Capability Scaling.** To examine how model capacity influences A2R, we evaluate the Qwen2.5-Deepseek-distilled series at 1.5B, 7B, and 32B. Results in Table 2 reveal a clear scaling trend. The 1.5B model falls short of its self-consistency (Cons@8) baseline, suggesting that a model with insufficient reasoning capacity struggles even to select the consensus answer in complex scenarios. The 7B model performs on par with self-consistency, suggesting that moderate ability allows A2R to match but not exceed the baseline. In contrast, the 32B model gains substantially, surpassing Cons@8 and approaching its Pass@8 limit. These findings indicate that A2R's advantages are unlocked only when the Synthesizer has strong reasoning capability.

**Asymmetric Configurations.** We further tested asymmetric allocations by swapping the roles of Qwen2.5-7B-D and Qwen2.5-32B-D. When the 32B model serves as Synthesizer for paths generated by the 7B Explorer, it achieves 77.92 on AIME24—surpassing not only the Explorer's Cons@K (65.17) and the Synthesizer's own Pass@1 (67.00), but even the Explorer's theoretical Pass@K bound (75.89). This demonstrates that a strong Synthesizer can actively re-reason over references rather than merely selecting them. In contrast, when the 7B model is used as Synthesizer, performance drops to 71.25, below its Cons@K baseline (78.95), highlighting that weaker Synthesizers cannot effectively exploit references. Since exploration dominates computational cost, a "small Explorer, large Synthesizer" setup emerges as both efficient and effective.

**Performance Constraint.** We further examined asymmetric settings to assess whether A2R's effectiveness is bounded by the Synthesizer's capability. Using Qwen3-8B-D as the Explorer, we tested three Synthesizers: base Qwen3-8B, an RL-enhanced variant, and base Qwen3-8B-D. As shown in Table 2, the weaker 8B Synthesizer yields poor results (85.21 vs. 86.27 for self-consistency), and RL fine-tuning offers only marginal recovery (86.87). By contrast, the stronger 8B-D Synthesizer delivers substantial gains, approaching its Pass@K upper bound. These findings confirm that the Synthesizer is the critical bottleneck: it must deeply re-reason over references rather

| Configuration (E → S) | Dataset | Pass@1 | Cons@8 | Pass@8 | A2R |
|---|---|---|---|---|---|
| *Capability Scaling – Qwen2.5 Family* | | | | | |
| 1.5B-D → 1.5B-D | AIME24 | 25.24 | 35.86 | 53.30 | 32.10 |
| | AIME25 | 22.10 | 29.62 | 37.43 | 25.00 |
| 7B-D → 7B-D | AIME24 | 50.19 | 65.17 | 75.89 | 65.85 |
| | AIME25 | 37.57 | 47.18 | 60.96 | 41.67 |
| 32B-D → 32B-D | AIME24 | 67.00 | 78.95 | 84.07 | 84.17 |
| | AIME25 | 53.56 | 64.31 | 75.26 | 70.84 |
| *Asymmetric Configurations – Qwen2.5 Family* | | | | | |
| 7B-D → 32B-D | AIME24 | 50.19 | 65.17 | 75.89 | 77.92 |
| | AIME25 | 37.57 | 47.18 | 60.96 | 57.45 |
| 32B-D → 7B-D | AIME24 | 67.00 | 78.95 | 84.07 | 71.25 |
| | AIME25 | 53.56 | 64.31 | 75.26 | 58.32 |
| *Performance Constrain – Qwen3 Family* | | | | | |
| 8B-D → 8B | AIME24 | 82.50 | 86.27 | 90.64 | 85.21 |
| | AIME25 | 74.59 | 81.44 | 86.70 | 79.64 |
| 8B-D → 8B (Opt) | AIME24 | 82.50 | 86.27 | 90.64 | 86.87 |
| | AIME25 | 74.59 | 81.44 | 86.70 | 81.47 |
| 8B-D → 8B-D | AIME24 | 82.50 | 86.27 | 90.64 | 89.59 |
| | AIME25 | 74.59 | 81.44 | 86.70 | 83.34 |

Table 2: The table details three experiments: (1) **Capability Scaling**, where Explorer and Synthesizer models are identical and scaled up; (2) **Asymmetric Configurations**, where smaller Explorer models are paired with larger Synthesizers and vice versa; and (3) **Performance Constrain**, which demonstrates the framework's ability to elicit the Synthesizer's latent capabilities. D represents the model is distilled from Deepseek.

than merely route answers, and A2R is most effective when combining an efficient Explorer with the most capable Synthesizer available.

## 4.4 **A2R-Efficient**: High Performance with Optimal Resource Allocation

Motivated by our analysis confirming that a powerful Synthesizer is the key to performance, we now present A2R-Efficient, an asymmetric framework designed for optimal resource allocation. This configuration utilizes a smaller, cost-effective model as the Explorer and a larger, more capable model as the Synthesizer, which is further optimized with reinforcement learning. We evaluate this framework to demonstrate that an intelligent allocation of computational resources can achieve performance comparable to or exceeding that of a much larger monolithic model, but with significantly greater efficiency. To reflect real-world costs, we calculate the total cost based on the official Qwen API pricing. Detailed pricing calculation and token usage is shown in the appendix A.1

A striking initial result from Table 3 is the power of A2R even on a small model. The symmetric Qwen3-4B A2R configuration achieves an average score of 67.80, effectively matching the performance of the monolithic Qwen3-32B model (67.76) at a 31% lower computational cost. This demonstrates that a small model can replicate the performance of a model 8x its size by leveraging the structured re-reasoning of the A2R framework, offering a highly efficient alternative to deploying massive models for baseline tasks.

Building on this, the asymmetric Qwen3-4B + Qwen3-8B configuration further validates our core principle of allocating greater resources to the critical synthesis stage. This pairing boosts the average score to 68.31, definitively surpassing the Qwen3-32B baseline. Crucially, this significant performance gain is achieved with only a minimal increase in computational overhead compared to the symmetric Qwen3-4B setup. This is because the total inference cost is dominated by the $N$ parallel rollouts of the Explorer stage, making the upgrade of the single-pass Synthesizer a highly efficient investment. This result shows that by adding a slightly larger model for synthesis—a computationally lighter task than parallel exploration—we can achieve performance superior to that of a much larger, single model.

| Benchmark | Configuration (E → S) | Pass@1 | Cons@4 | Pass@4 | A2R | Cost/1K |
|---|---|---|---|---|---|---|
| AIME 2024 | 4B → 4B | 74.63 | 79.38 | 81.44 | 80.03 | 0.201 |
| | 4B → 8B | 74.63 | 79.38 | 81.44 | 81.13 | 0.212 |
| | 4B → 8B(Opt) | 74.63 | 79.38 | 81.44 | 80.63 | 0.211 |
| | 8B (Baseline) | 75.30 | 79.34 | 82.76 | — | 0.078 |
| | 32B (Baseline) | 81.15 | 84.74 | 88.75 | — | 0.271 |
| AIME 2025 | 4B → 4B | 65.83 | 73.70 | 80.47 | 74.44 | 0.245 |
| | 4B → 8B | 65.83 | 73.70 | 80.47 | 74.28 | 0.257 |
| | 4B → 8B(Opt) | 65.83 | 73.70 | 80.47 | 76.70 | 0.251 |
| | 8B (Baseline) | 68.67 | 74.84 | 80.48 | — | 0.097 |
| | 32B (Baseline) | 73.32 | 78.45 | 84.14 | — | 0.341 |
| BeyondAIME | 4B → 4B | 42.10 | 45.75 | 55.39 | 48.94 | 0.242 |
| | 4B → 8B | 42.10 | 45.75 | 55.39 | 49.51 | 0.253 |
| | 4B → 8B(Opt) | 42.10 | 45.75 | 55.39 | 50.26 | 0.254 |
| | 8B (Baseline) | 43.94 | 47.24 | 57.13 | — | 0.101 |
| | 32B (Baseline) | 48.80 | 52.39 | 61.66 | — | 0.365 |
| Average | 4B → 4B | 60.85 | 66.28 | 72.43 | 67.80 | 0.235 |
| | 4B → 8B | 60.85 | 66.28 | 72.43 | 68.31 | 0.246 |
| | 4B → 8B(Opt) | 60.85 | 66.28 | 72.43 | 69.20 | 0.245 |
| | 8B (Baseline) | 62.64 | 67.14 | 73.46 | — | 0.092 |
| | 32B (Baseline) | 67.76 | 71.86 | 78.18 | — | 0.343 |

Table 3: Comparison of computational costs for different size of Qwen3 model and A2R framework configurations, measured in cost per thousand tokens.

The full potential of A2R-Efficient is unlocked through reinforcement learning. The final Qwen3-4B + Qwen3-8B(Opt) configuration achieves the highest overall average score of 69.20, setting a new performance benchmark at a 29% lower computational cost. Additionally, due to the synthesizer's concise output, this approach does not introduce significant user-facing latency, making it highly practical for real-world applications.

## 4.5 ABLATION

In this section, we investigate key settings that influence the stability of our reinforcement learning training. Specifically, we examine the impact of on-policy versus off-policy update strategies and the effect of temperature on policy entropy.

### 4.5.1 UPDATE STATERY

We investigate the impact of on-policy versus off-policy update strategies on training stability. For this analysis, we use a baseline Reinforcement Learning with Verifiable Rewards (RLVR) setup, training the Qwen3-4B model on a math domain dataset with the DAPO algorithm Yu et al. (2025). A key modification was made to the training configuration: the 'clip-higher' value was deliberately set to match the 'clip-low' value. To compare the two approaches, the on-policy setting was implemented by making the mini-batch size equal to the full batch size. Conversely, the off-policy setting was established by making the full batch size four times larger than the mini-batch size.

As shown in Figure 3, we observe that while the off-policy setting achieves a rapid reward increase due to more frequent policy updates, this approach also causes the policy entropy to rise sharply. This escalating entropy leads to a concurrent increase in the gradient norm and a steep decline in response length, which indicates significant training instability. In contrast, the on-policy setting, despite exhibiting a slower initial reward increase, maintains much more stable training dynamics across other key metrics. This stability permits longer and more consistent training, ultimately leading to superior results. Therefore, we adopt the on-policy strategy as the default configuration for all subsequent experiments.

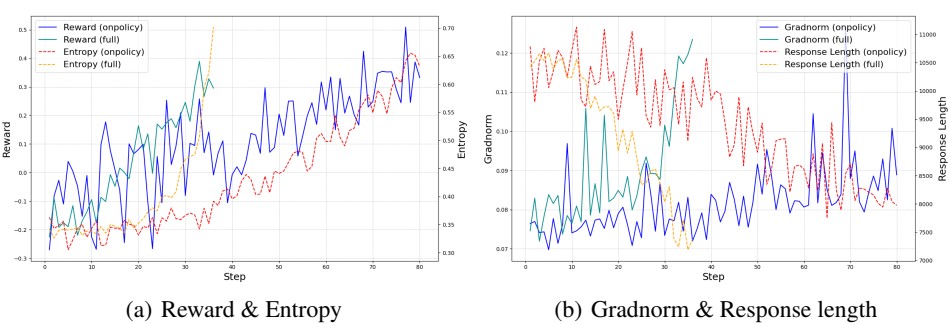

(a) Reward & Entropy                    (b) Gradnorm & Response length

Figure 2: On-Policy vs. Off-Policy Training Dynamics

### 4.5.2 TEMPERATURE CONTROLLED

We found that the general temperature setting of 1.0 leads to training collapse at approximately 250 steps. Although the policy entropy increases steadily before step 200, it subsequently accelerates rapidly to an extremely high value, which coincides with a significant drop in model performance. Motivated by this instability, we investigated the influence of different temperature settings on training dynamics. Keeping all other hyperparameters identical, we conducted a comparative experiment with the temperature set to 0.7. We observed that with this lower temperature, the policy entropy starts from a lower initial point and increases more slowly over time. This configuration not only produces a more stable response length curve compared to the standard setting but also achieves a higher performance upper bound. Consequently, we adopted a temperature of 0.7 as the optimal setting for our experiments.

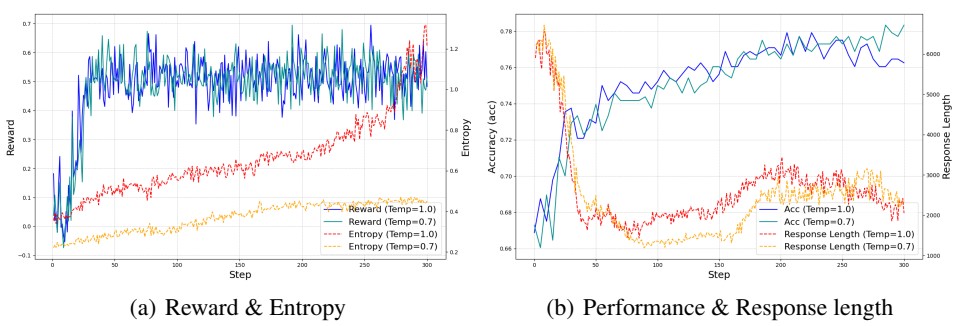

(a) Reward & Entropy                    (b) Performance & Response length

Figure 3: High-Temperature vs. Low-Temperature Training Dynamics

## 5 CONCLUSION

In this work, we introduced **Asymmetric Two-Stage Reasoning (A2R)**, a framework that explicitly decouples inference into exploration and synthesis phases. By enabling a Synthesizer model to perform generative re-reasoning over the diverse outputs of an Explorer, **A2R** substantially narrows the gap between a model's realized and latent reasoning capabilities. Our experiments demonstrate consistent improvements over self-consistency baselines across multiple reasoning benchmarks, validating that active synthesis is more powerful than passive aggregation. Through systematic analysis, we showed that the Synthesizer's intrinsic capability is the critical determinant of overall performance. Stronger synthesizers not only yield greater improvements but also unlock the latent potential of weaker explorers, confirming the necessity of deep re-reasoning rather than simple answer selection. Furthermore, our proposed asymmetric "small Explorer, large Synthesizer" configuration achieves performance on par with much larger monolithic models while reducing computation cost by nearly 30%, offering a practical and efficient deployment strategy.

## ETHICS STATEMENT

The authors of this paper have read and adhered to the ICLR Code of Ethics. Our research focuses on a foundational inference-time paradigm, Parallel Reasoning, aimed at improving the performance and computational efficiency of language models. The work is algorithmic in nature and does not involve the use of human subjects or personally identifiable information.

## REPRODUCIBILITY STATEMENT

We are committed to ensuring the reproducibility of our research. To this end, the complete source code for our Parallel Reasoning framework, including all scripts for data processing, evaluation, and replication of our experimental results will be provided in an anonymous GitHub repository. All of our experiments were conducted on publicly available benchmarks. A comprehensive description of these datasets, our experimental setup, model hyperparameters, and specific evaluation protocols is detailed in 4.1. Furthermore, all mathematical derivations and complete proofs for our theoretical claims regarding computation allocation can be found in Appendix A.1. We believe these resources provide a clear and direct path for replicating our findings.

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

# A    APPENDIX

## A.1    TOKEN USAGE AND PRICING

| Benchmarks | Explorer | | | Synthesizer | | | Metric | | | | Cost/1K |
|---|---|---|---|---|---|---|---|---|---|---|---|
| | Model | Input Len | Output Len | Model | Input Len | Output Len | Pass@1 | Cons@4 | Pass@4 | A2R | |
| AIME 2024 | 4B | 105 | 15768 | 4B | 4577 | 4796 | 74.63 | 79.38 | 81.44 | 80.03 | 0.201 |
| | 4B | 105 | 15768 | 8B | 4577 | 4879 | 74.63 | 79.38 | 81.44 | 81.13 | 0.212 |
| | 4B | 105 | 15768 | 8B(Opt) | 4577 | 4428 | 74.63 | 79.38 | 81.44 | 80.63 | 0.211 |
| | 8B | 105 | 15895 | - | - | - | 75.30 | 79.34 | 82.76 | - | 0.078 |
| | 32B | 105 | 13783 | - | - | - | 81.15 | 84.74 | 88.75 | - | 0.271 |
| AIME 2025 | 4B | 159 | 19205 | 4B | 4610 | 5808 | 65.83 | 73.70 | 80.47 | 74.44 | 0.245 |
| | 4B | 159 | 19205 | 8B | 4610 | 5807 | 65.83 | 73.70 | 80.47 | 74.28 | 0.257 |
| | 4B | 159 | 19205 | 8B(Opt) | 4610 | 4574 | 65.83 | 73.70 | 80.47 | 76.70 | 0.251 |
| | 8B | 159 | 19744 | - | - | - | 68.67 | 74.84 | 80.48 | - | 0.097 |
| | 32B | 159 | 17286 | - | - | - | 73.32 | 78.45 | 84.14 | - | 0.341 |
| BeyondAIME | 4B | 129 | 19263 | 4B | 4293 | 4622 | 42.10 | 45.75 | 55.39 | 48.94 | 0.242 |
| | 4B | 129 | 19263 | 8B | 4293 | 4766 | 42.10 | 45.75 | 55.39 | 49.51 | 0.253 |
| | 4B | 129 | 19263 | 8B(Opt) | 4293 | 5032 | 42.10 | 45.75 | 55.39 | 50.26 | 0.254 |
| | 8B | 129 | 20553 | - | - | - | 43.94 | 47.24 | 57.13 | - | 0.101 |
| | 32B | 129 | 18554 | - | - | - | 48.80 | 52.39 | 61.66 | - | 0.365 |
| Average | 4B | 130 | 18597 | 4B | 4406 | 4884 | 60.85 | 66.28 | 72.43 | 67.80 | 0.235 |
| | 4B | 130 | 18597 | 8B | 4406 | 4982 | 60.85 | 66.28 | 72.43 | 68.31 | 0.246 |
| | 4B | 130 | 18597 | 8B(Opt) | 4406 | 4832 | 60.85 | 66.28 | 72.43 | 69.20 | 0.245 |
| | 8B | 130 | 18731 | - | - | - | 62.64 | 67.14 | 73.46 | - | 0.092 |
| | 32B | 130 | 17422 | - | - | - | 67.76 | 71.86 | 78.18 | - | 0.343 |

Table 4: Comparison of computational costs for different size of Qwen3 model and A2R framework configurations, measured in cost per thousand tokens.

We calculated the total cost using the following formula:

$$\text{Cost}_{\text{Explorer}} = N \times (T_{\text{in,E}} \times P_{\text{in}} + T_{\text{out,E}} \times P_{\text{out}}) \qquad (2)$$

$$\text{Cost}_{\text{Synthesizer}} = T_{\text{in,S}} \times P_{\text{in}} + T_{\text{out,S}} \times P_{\text{out}} \qquad (3)$$

$$\text{Cost}_{\text{Total}} = \text{Cost}_{\text{Explorer}} + \text{Cost}_{\text{Synthesizer}} \qquad (4)$$

where we need to precisely measure the number of input and output tokens (represented by $T_{\text{in}}$ and $T_{\text{out}}$, respectively) and apply their separate prices to accurately calculate the final cost.

## A.2    THE USE OF LARGE LANGUAGE MODELS (LLMS)

In accordance with the conference policy, we disclose the use of Google's Gemini (Comanici et al., 2025) large language model (LLM) as an assistive tool in the preparation of this manuscript. The LLM's primary role was as an advanced writing assistant, used for tasks such as rephrasing sentences for improved clarity, correcting grammatical errors, and polishing the academic tone. Specific sections, including the introduction, experimental methods, and the reproducibility statement, were iteratively refined with the model's assistance to enhance their precision and readability. While the LLM provided significant assistance with language and drafting, the core research ideas, experimental design, and scientific contributions are entirely the work of the human authors. The authors maintained full intellectual control over the content and bear full responsibility for all claims and any potential errors in the final manuscript.

