# OpenReview forum: "A2R: An Asymmetric Two-Stage Reasoning Framework for Parallel Reasoning"
_ICLR.cc/2026/Conference — Submitted to ICLR 2026_

### Official Review · Reviewer_rhPN · 2025-10-24

**Soundness:** 2
**Presentation:** 3
**Contribution:** 2
**Rating:** 4
**Confidence:** 5

**Summary:**

This paper presents A2R, a novel two-stage reasoning framework that uses an "Explorer" model to generate multiple reasoning paths in parallel and a "Synthesizer" model to perform generative re-reasoning over these paths, significantly boosting performance on complex reasoning tasks and introducing an efficient asymmetric scaling paradigm. The authors further introduce A2R-Efficient, an asymmetric variant pairing a small Explorer with a larger, RL-fine-tuned Synthesizer, that matches or exceeds the performance of a much larger model.

**Strengths:**

(1) Clarity and Execution: The paper is well-written and the framework is clearly explained. The experimental evaluation is rigorous, comprehensive across multiple benchmarks (AIME, BeyondAIME), and the results are solid.

(2) Systematic Analysis: The analysis dissecting the Synthesizer's role is thorough and provides valuable insights, clearly identifying the Synthesizer's capacity as the critical performance driver.

(3) Practical Contribution: The proposed "A2R-Efficient" paradigm and the empirical result that an asymmetric setup can match a much larger monolithic model at a lower cost is a practical and potentially useful finding for deployment.

**Weaknesses:**

(1) Lack of Novelty: The fundamental two-stage, generate-and-consolidate paradigm is similar to prior work [1]. The paper does not make a compelling case for why A2R represents a significant conceptual shift rather than a specific implementation of an existing idea.

(2) Limited Analysis of Exploration Diversity: The framework relies on the Explorer generating "diverse" reasoning paths, but the paper does not quantify or actively control for this diversity, leaving a key component of the method's success unexplained.

Reference:
[1] Improving Factuality and Reasoning in Language Models through Multiagent Debate

**Questions:**

(1) How do you define and ensure "diverse" reasoning paths from the Explorer? Did you experiment with techniques to explicitly maximize diversity and measure its correlation with the Synthesizer's performance?

(2) The paper primarily evaluates A2R against self-consistency and pass@K baselines. Have the authors considered comparing it with other synthesis-based approaches?

---

### Official Review · Reviewer_wtaF · 2025-10-27

**Soundness:** 3
**Presentation:** 3
**Contribution:** 3
**Rating:** 4
**Confidence:** 3

**Summary:**

This paper introduces A2R, an Asymmetric Two-Stage Reasoning framework designed to bridge the gap between a model’s single-pass reasoning performance and its latent multi-path potential. The method decomposes inference into two distinct stages: (1) an Explorer that generates multiple reasoning paths in parallel, and (2) a Synthesizer that integrates these paths through generative re-reasoning. The authors further identify that the Synthesizer’s reasoning capability is the critical performance bottleneck, leading to the proposal of an asymmetric configuration (A2R-Efficient) — a small Explorer combined with a large, RL-enhanced Synthesizer. Experiments on challenging mathematical reasoning benchmarks (AIME 2024, AIME 2025, BeyondAIME) show substantial gains over self-consistency and single-pass baselines, with up to 75% relative improvement and ~30% reduced computational cost compared to larger monolithic models.

**Strengths:**

Novel and well-motivated framework.
The idea of explicitly separating exploration and synthesis within inference-time reasoning is both conceptually elegant and practically meaningful. It generalizes beyond simple voting or ranking methods used in self-consistency.

Strong empirical validation.
The experiments are extensive and demonstrate clear, consistent improvements across multiple benchmarks and model scales, including ablations and scaling analyses that support the claims.

Insightful analysis of asymmetry.
The paper provides a detailed examination of the roles of Explorer and Synthesizer, revealing that asymmetric scaling (small Explorer, large Synthesizer) is not only efficient but also theoretically sound. The efficiency–performance trade-off analysis is convincing

**Weaknesses:**

The experiments in this paper are not sufficiently comprehensive, as they only use the Qwen3 series models. No larger-scale or closed-source models were included for validation.

The evaluation in this paper also appears to focus mainly on the AIME dataset, without including other mathematical benchmarks.

**Questions:**

None

---

### Official Review · Reviewer_Y8gM · 2025-10-29

**Soundness:** 3
**Presentation:** 4
**Contribution:** 3
**Rating:** 4
**Confidence:** 3

**Summary:**

The authors argue that standard multi-path methods, like self-consistency, use a "passive aggregation" (voting) that fails to leverage the rich information within different reasoning paths. To solve this, the paper proposes A2R, an Asymmetric Two-Stage Reasoning framework. In this framework, a first-stage "Explorer" model generates N diverse reasoning paths in parallel. Then, a second-stage "Synthesizer" model receives these paths as references and performs a "generative re-reasoning" to produce a single, refined answer.

The paper's key insight is that the Synthesizer's reasoning capability is the primary performance bottleneck. This leads to the main contribution: A2R-Efficient, a "small-to-big" asymmetric architecture where a small, cheap Explorer is paired with a large, powerful Synthesizer. Experiments show this asymmetric setup (e.g., a 4B Explorer + 8B Synthesizer) can outperform a much larger monolithic 32B model, while reducing computational costs by nearly 30%.

**Strengths:**

1. The discovery that the Synthesizer's capability is the critical bottleneck is a key insight that has not been well-explored. The "small-to-big" A2R-Efficient architecture is a highly practical and effective outcome of this insight. The ability to achieve superior performance to a 32B model at nearly 30% lower cost (Table 3) is a very strong and significant result for the community.
2. The authors systematically validate their claims by first showing A2R's general effectiveness (Table 1), then isolating the critical component (Table 2), and finally demonstrating a cost-effective application (Table 3).

**Weaknesses:**

1. Synthesizer only receives the answer components from the Explorer paths, not the full reasoning traces, due to context length. This seems like a significant limitation. The model might miss crucial context if an answer is correct but the reasoning is flawed, or vice-versa. This design choice feels under-discussed.
2. The cost analysis is based on API pricing (total tokens), which is a good proxy but not a complete picture. The A2R framework is sequential (Stage 1, then Stage 2). This adds latency. It's unclear if the A2R-Efficient (4B -> 8B) model, while 29% cheaper in tokens, is actually faster.
3. The experiments are primarily focused on the Qwen model family and the domain of mathematical reasoning (AIME). It would strengthen the paper to show that this "asymmetric scaling" benefit also applies to other model architectures (e.g., Llama, Mistral) and other reasoning domains (e.g., code generation, logical puzzles).

**Questions:**

The cost savings in Table 3 are compelling. However, have you measured the end-to-end wall-clock latency? How does the latency of the two-stage A2R-Efficient (e.g., 4B Explorer + 8B Synthesizer) compare to the single-stage 32B baseline model? Is there a speed-cost trade-off, or is A2R-Efficient better on both metrics?

---

### Official Review · Reviewer_VrYT · 2025-11-05

**Soundness:** 2
**Presentation:** 3
**Contribution:** 2
**Rating:** 2
**Confidence:** 3

**Summary:**

This paper introduces A2R, a two-stage framework where a "explorer" model first generates multiple reasoning paths, and then a "synthesizer" model performs a generative re-reasoning over these paths to produce a superior final answer. And A2R-Efficient,  which uses a lightweight Explorer with a stronger Synthesize, this asymmetric approach achieves performance comparable to a large model at a lower computational cost.

**Strengths:**

1. This paper proposes a two-stage framework that performs parallel reasoning using smaller models, followed by re-reasoning with a larger model. This approach is reasonable.
2. The paper conducts experiments across multiple model sizes and datasets, exploring different model combinations to build a more efficient asymmetric framework that achieves good results while reducing computational cost.
3. The details of the RL experiments are clearly described.

**Weaknesses:**

1. The improvement in experimental results is minimal. In Table 1, the average gain over Cons@N is at most 2.76, and only 1.52 for the 4B model. However, since the AIME dataset contains only 30 questions, this improvement corresponds to less than one additional correct answer.
2. The necessity of RL in this paper’s experiments is questionable. In this framework, the “synthesizer” can naturally use the “explorer” responses as supplementary information, without requiring RL to learn this behavior. Moreover, aside from the original reward, the paper does not design a reward function aligned with the proposed framework. As a result, the simple RL training brings less than a 1% improvement in Table 3 (from 68.31% to 69.20%), indicating very limited necessity.
3. The paper lacks deeper analysis. Although it claims to perform “re-reasoning” by generating multiple reasoning paths with one model and summarizing them with another, the approach remains rather naive. More analysis is needed to demonstrate the broader advantages of this design.

**Questions:**

1. Could you provide an example showing how the “synthesizer” considered different responses and ultimately arrived at the correct answer?

---

### Meta-Review · Area_Chair_Bmo1 · 2025-12-30

**Summary:**

Reviewers’ core concerns include: minimal experimental improvement (especially on small datasets), questionable necessity of RL in the framework, insufficient analysis of the re-reasoning design; limited generality (over-reliance on Qwen models and AIME dataset), incomplete cost analysis lacking latency measurement; insufficient novelty compared to prior two-stage paradigms, and unquantified/undermanaged diversity of Explorer-generated reasoning paths.

**Reviewer Concerns:**

None, as no rebuttal was provided

**Reviewer Scores:**

N/A

---

### Decision · Program_Chairs · 2026-01-26

Reject